# Prevalence and Individualized Risk Factors of *E. bieneusi* and *E. intestinalis* Infections Among People Living with HIV (PLHIV) with Diarrhea in Ecuador: Insights from a Single-Center Cross-Sectional Study

**DOI:** 10.3390/jcm14020348

**Published:** 2025-01-08

**Authors:** Betty J. Pazmiño-Gómez, Jennifer Rodas-Pazmiño, Fabricio Guevara-Viejó, Karen Rodas-Pazmiño, Roberto Coello-Peralta, Edgar Rodas-Neira, Marco Faytong-Haro, Luis Cagua-Montaño

**Affiliations:** 1Universidad Estatal de Milagro, Milagro 091706, Ecuador; jguevarav@unemi.edu.ec (F.G.-V.); krodasp2@unemi.edu.ec (K.R.-P.); lcaguam@unemi.edu.ec (L.C.-M.); 2Laboratorio Clínico y Microbiológico Pazmiño, Milagro 091706, Ecuador; jrodas@labpazmino.com (J.R.-P.); erodas@labpazmino.com (E.R.-N.); 3Departamento de Microbiología, Facultad de Medicina Veterinaria y Zootecnia, Universidad de Guayaquil, Guayaquil 090514, Ecuador; roberto.coellope@ug.edu.ec

**Keywords:** antiretroviral therapy adherence, Ecuador, *E. bieneusi*, *E. intestinalis*, microsporidia, people living with HIV, PLHIV, prevalence, risk factors

## Abstract

**Background**: Microsporidia, particularly *E. bieneusi* and *E. intestinalis*, are emerging opportunistic pathogens that pose significant health risks to immunocompromised individuals, especially people living with HIV (PLHIV). Despite the global recognition of microsporidia’s impact, there has been limited research on their prevalence and associated risk factors in Ecuador. This study aimed to investigate the prevalence and identify risk factors associated with microsporidia infections among PLHIV with diarrhea in Ecuador. **Methods**: A cross-sectional study was conducted at the José Daniel Rodríguez Infectious Hospital in Guayaquil, Ecuador, between April 2021 and May 2022. A total of 85 PLHIV with diarrhea were included in the analysis. Fecal samples were tested for microsporidia using fast-hot Gram chromotrope staining, immunofluorescence microscopy, and transmission electron microscopy. Logistic regression was performed to assess the association between demographic, clinical, and behavioral factors and microsporidia infection. **Results:** The prevalence of microsporidia infections among the study population was 18.8%. Logistic regression analysis identified significant associations with age ≥ 60 years (OR = 19.169, *p* = 0.036), female sex (OR = 10.491, *p* = 0.028), and non-adherence to antiretroviral therapy (OR = 8.466, *p* = 0.077). Marginally significant associations were observed for substance use (OR = 0.262, *p* = 0.094) and high/very high HIV viral load (≥10,000 copies/mL, OR = 0.457, *p* = 0.094). CD4 count and marital status showed descriptive trends but were not statistically significant. **Conclusions:** This study highlights the burden of microsporidia infections among PLHIV in Ecuador and underscores the need for targeted public health interventions. Strategies should prioritize older individuals, females, and those with poor ART adherence to reduce infection risk. Future research is needed to explore additional risk factors and refine precision medicine approaches for this population.

## 1. Introduction

As global health challenges continue to evolve, the need for personalized approaches in medicine has become increasingly evident, particularly in the management of HIV/AIDS [1,2,3]. Understanding the spread and impact of opportunistic infections like microsporidia among PLHIV is especially critical in regions with developing healthcare systems [4], where personalized medicine can play a pivotal role in improving patient outcomes and reducing the burden of these infections. Microsporidia, emerging as significant pathogens, pose substantial health risks to immunocompromised individuals, especially PLHIV [5]. These patients are particularly vulnerable due to the depletion of CD4+ lymphocytes, which weakens their immune systems and heightens their susceptibility to opportunistic infections such as severe diarrhea. Initially misclassified as protozoa, microsporidia were later recognized as fungi and obligate intracellular parasites, infecting a wide range of hosts, including both vertebrates and invertebrates. Since their discovery in silkworms by Naegeli in 1857 [6,7], microsporidia have gained recognition for their substantial impact on human health.

In the context of personalized medicine, understanding the individual risk factors and characteristics of microsporidia infections among PLHIV is crucial for developing tailored therapeutic strategies [8]. The first human microsporidia species, *Enterocytozoon bieneusi*, was identified in 1985 in Haiti among PLHIV with watery diarrhea. This species remains the primary cause of microsporidiosis in humans. Another significant species, *Septata intestinalis*, now classified within the genus *Encephalitozoon*, is also a major contributor to disseminated diarrhea among these patients [9,10,11].

Microsporidia infections have been documented worldwide, with prevalence rates among diarrheal patients varying significantly by region. In Europe and the United States, reported infection rates range from 12% to 50%. Prevalence rates from other parts of the world include 36% in Germany, 7% in Niger Bretagne, 10% in Zimbabwe [12], 7.3% in Spain [13], 1.4% in China [14], 31% in Mexico, 27% in Brazil, and 3.5% in Colombia [15,16]. In Africa, studies report prevalence rates of 9.4% in Senegal [17] and 13.6% in Venezuela [18]. In Ecuador, the reported prevalence among PLHIV with diarrhea is an alarming 25%, suggesting that microsporidia may be a significant, yet underexplored, public health issue in this region [19].

Advancements in omics technologies, such as genomics and proteomics, offer new opportunities to explore the host–pathogen interactions in microsporidia infections, potentially leading to more precise diagnostic and therapeutic interventions that are tailored to the genetic and molecular profiles of individual patients [20]. Despite the global recognition of microsporidia’s impact, there has been no comprehensive study on their prevalence and characteristics in Ecuador, leaving a critical gap in our understanding of their significance in this context [21]. The absence of detailed epidemiological data on microsporidia in Ecuador limits the development of effective public health interventions and tailored treatment strategies.

The use of advanced diagnostic techniques, such as immunofluorescence microscopy and transmission electron microscopy, not only allows for accurate detection of microsporidia but also lays the groundwork for the development of personalized diagnostic tools that can be adapted to the unique characteristics of different patient populations [22,23]. Additionally, conducting research in vulnerable populations, such as PLHIV, requires careful consideration of ethical principles, particularly in the context of personalized medicine. Ensuring informed consent, maintaining patient confidentiality, and addressing the unique healthcare needs of these patients are critical components of ethical research practice that can also enhance the effectiveness of personalized treatment strategies [24,25,26].

This study is the first to systematically investigate the prevalence and characteristics of microsporidia infections among PLHIV with diarrhea in Ecuador. By analyzing clinical, sociodemographic, and laboratory data, this research aims to quantify the prevalence of microsporidia in this population, identify associated risk factors, and explore the public health implications of these findings. While this study focuses on the PLHIV population in Ecuador, the findings have broader implications for personalized medicine in similar epidemiological settings. By filling this gap, the study will provide valuable insights that can inform public health strategies and improve the management of microsporidia infections, not only in Ecuador but also in other regions with similar epidemiological profiles. Additionally, this research will contribute to the global body of knowledge on microsporidia, offering data that may be beneficial for comparative studies and future investigations.

This research not only addresses a critical gap in the epidemiological understanding of microsporidia infections in Ecuador but also aligns with the broader goals of precision medicine. By identifying patient-specific risk factors and leveraging advanced diagnostic technologies, this study paves the way for more personalized approaches to managing microsporidia infections, ultimately improving patient care and outcomes.

## 2. Materials and Methods

### 2.1. Data Sources and Study Population

This study was conducted at the José Daniel Rodríguez Infectious Hospital in the City of Guayaquil, Ecuador. The study population consisted of PLHIV with diarrheal syndrome who met the following inclusion criteria: (1) confirmed diagnosis of HIV; (2) age 18 years or older; (3) presence of diarrheal symptoms for at least 7 days; and (4) willingness to provide informed consent and participate in the study. Exclusion criteria included (1) any other confirmed cause of diarrhea unrelated to HIV or microsporidia infection, such as bacterial or parasitic infections identified through preliminary stool testing; (2) recent use of antibiotics or antifungal treatments within the past 4 weeks; and (3) refusal or inability to provide informed consent. A total of 92 patients were initially enrolled, with 85 meeting the final criteria for inclusion in the analysis.

The study was designed as a descriptive, correlational, non-experimental, observational study with a cross-sectional, prospective design. Clinical histories were collected for each participant, providing comprehensive data on their medical background and current health status. All participants provided informed consent, and their confidentiality was assured throughout the study. Clinical, sociodemographic, and laboratory data were collected for each patient using a standardized data collection sheet. The study was approved by the Human Ethics Committee of the Hospital Luís Vernaza Guayaquil-Ecuador HLV-CEISH-2021-011 on 05-03-2021.

### 2.2. Sample Collection and Laboratory Analysis

Fecal samples were collected from participating PLHIV in sterile containers to prevent contamination and immediately placed in cold storage at 4 °C to maintain sample integrity during transport to the National Institute for Public Health and Research (INSPI). Upon arrival at the laboratory, samples were promptly processed for light microscopy with Chromotropic Gram staining and for immunofluorescence microscopy. To allow for subsequent analyses, a portion of each sample was preserved in 10% formalin and stored for later examination using transmission electron microscopy.

To detect microsporidia, the samples were stained using Chromotropic Gram staining and immunofluorescence techniques. For Chromotropic Gram staining, samples were fixed with absolute methanol for 5 min, stained with gentian violet for 1 min, and then washed. Samples were subsequently treated with Lugol’s iodine, decolorized with alcohol, and counterstained with Chromotropo 2R solution, which allowed for the identification of violet-stained spores under light microscopy at 1000× magnification. For further confirmation, immunofluorescence staining was performed using monoclonal antibodies specific to *E. bieneusi* and *E. intestinalis*. After incubation and washing with PBS, slides were treated with a fluorescent anti-mouse IgG conjugate and observed under a fluorescence microscope equipped with a 488 nm filter at 1000× magnification. Microsporidia were identified using Chromotropic Gram stain, as shown in Figure 1.

For samples that tested positive for microsporidia, further analysis was conducted using immunofluorescence microscopy with specific monoclonal antibodies to detect *E. bieneusi* and *E. intestinalis*. The presence of *E. bieneusi* was confirmed using immunofluorescence with monoclonal antibodies (Figure 2).

Similarly, *E. intestinalis* was detected through immunofluorescence with monoclonal antibodies (Figure 3).

Additionally, transmission electron microscopy was employed to observe the ultra-internal structures of the microorganisms, which facilitated their differentiation [27,28]. You can see one of our samples in Figure 4 and Figure 5.

### 2.3. Variables

#### 2.3.1. Outcome Variable

*E. bieneusi* or *E. intestinalis* Status: The primary outcome variable was the presence or absence of *E. bieneusi* or *E. intestinalis* infection, determined through laboratory analysis. Infections were identified using Gram chromotrope staining, immunofluorescence with species-specific monoclonal antibodies, and transmission electron microscopy. The prevalence of *E. bieneusi* was 16.5% (14 out of 85) and *E. intestinalis,* 2.4% (2 out of 85). Due to the sample size, we categorized patients as either positive or negative for these microsporidia species based on the presence of infection by either *E. bieneusi* or *E. intestinalis*. Although *E. bieneusi* and *E. intestinalis* are distinct species, they both cause similar gastrointestinal symptoms in immunocompromised individuals, particularly PLHIV, and were therefore combined for analysis.

Given our limited sample size, with only 16 total cases of infection, we did not meet the recommended threshold of 10 events per predictor variable in logistic regression [29], which would ideally require a minimum of 90 positive cases for our 9 predictors. By combining the two species, we aimed to maximize the available data and improve the stability of the model as much as possible. Despite this approach, the results should be interpreted with caution, and future studies with larger sample sizes could better explore species-specific risk factors.

#### 2.3.2. Predictor Variables

Age: The age of the patients was recorded and grouped into three categories, 18–39 years, 40–59 years, and 60+ years, reflecting distinct demographic and clinical stages. These categorizations were selected to explore age-related trends and their potential impact on the likelihood of microsporidia infection.

Sex: Biological sex affects immune responses and HIV progression. Studies have shown that women often have lower viral loads and higher CD4+ T-cell counts compared to men during early HIV infection, yet may experience faster disease progression. These differences are attributed to variations in immune activation and hormonal influences [30].

Marital Status: Marital status can impact health outcomes in people living with HIV (PLHIV). Research indicates that married individuals often have better mental health and higher adherence to antiretroviral therapy (ART) compared to unmarried individuals. This is likely due to increased social support and economic stability associated with marriage [31].

Educational Level: Education level was dichotomized into “Primary or Lower” and “Secondary or Higher” to assess whether educational attainment influenced infection rates.

Adherence to Antiretroviral Therapy (ART): ART adherence was categorized into three mutually exclusive groups based on a questionnaire administered at the time of sample reception. Participants who had been on ART for more than 30 days and self-reported missing doses for seven or more consecutive days in the past month were classified as non-adherent. Those who had been on ART for more than 30 days and reported consistent adherence without any interruptions in the past month were classified as adherent. Participants who initiated ART within the last 30 days, regardless of their adherence level, were classified as recently started. Self-reported responses were cross-checked against clinical records, when available, to enhance data accuracy.

HIV Viral Load: The viral load was measured in copies/mL, log-transformed (log10) for analysis, and treated as a continuous variable. This transformation provides easier-to-interpret values and was used to evaluate the association between viral load and the presence of microsporidia infection. For descriptive purposes, viral load was categorized into three groups—Undetectable/Low (<1000 copies/mL), Moderate (1000–9999 copies/mL), and High/Very High (≥10,000 copies/mL)—reflecting clinically meaningful thresholds commonly used in HIV research.

CD4 Count: The CD4+ T lymphocyte count was recorded as a continuous variable to assess its relationship with infection status. CD4 count is a key indicator of immune function in PLHIV. For categorical analysis, CD4 counts were grouped into three clinically relevant categories: Severely Low (<200 cells/mm^3^), Moderate (200–349 cells/mm^3^), and Normal/High (≥350 cells/mm^3^). These thresholds were chosen based on standard clinical practice to reflect varying levels of immune suppression.

Employment Status: Employment was recorded as a binary variable (employed or unemployed) to investigate its potential impact on infection risk.

Substance Use: Substance use was defined as the self-reported consumption of alcohol, tobacco, heroin, marijuana, cocaine, or crack during the last month. This variable was recorded as a binary response (yes or no) to explore its association with the presence of microsporidia infection. Participants who reported using any of these substances at least once in the past month were categorized as “yes”, while those who reported no usage were categorized as “no”.

Given the relatively small sample size and the exploratory nature of the study, we decided to collapse the individual substance categories into a single binary variable to ensure sufficient statistical power for the analysis and to simplify the interpretation of results. We acknowledge that this approach does not distinguish between sporadic use and dependency or addiction, which could influence the observed association. Furthermore, the original questionnaire was designed to be as concise as possible due to time constraints at the point of sample collection. As a result, detailed questions about the frequency or patterns of substance use were not included. Future studies with larger sample sizes and more detailed substance use assessments are recommended to explore the potential differential effects of substance use patterns on microsporidia infection.

### 2.4. Statistical Analysis

Descriptive and inferential statistical analyses were conducted using Microsoft Excel for data gathering and STATA 18.0 for data processing. Descriptive statistics were used to summarize the prevalence of *E. bieneusi* and *E. intestinalis* and to characterize the demographic and clinical features of the study population. Percentages and frequencies were calculated and presented in Table 1, which provides a detailed breakdown of the study population segmented by microsporidia infection status (*E. bieneusi* or *E. intestinalis* positive vs. negative) and totals. This segmentation enables a clear comparison of demographic and clinical characteristics between the infected and uninfected groups. Key variables include age categories, sex, marital status, education level, ART adherence, viral load, CD4 count, employment status, and substance use. Additionally, *p*-values from Chi-square are included to evaluate associations with infection status.

Additionally, a logistic regression model (Table 2) was used to assess the association between various epidemiological and clinical factors and the presence of microsporidia infection (*E. bieneusi* or *E. intestinalis*). The outcome variable was the presence of either *E. bieneusi* or *E. intestinalis*. Predictors included age (categorized into 18–39, 40–59, and 60+ years), sex, marital status, education level, ART adherence, viral load (categorized as undetectable/low, moderate, and high/very high), CD4 count (categorized as severely low, moderate, and normal/high), employment status, and substance use. All predictors were included simultaneously in a single multiple logistic regression model to account for potential confounding and interrelationships between predictors. The model provided odds ratios (ORs), 95% confidence intervals (CIs), and *p*-values to evaluate the strength and statistical significance of the associations. A *p*-value of less than 0.05 was considered statistically significant, while results with a *p*-value less than 0.10 were considered marginally significant.

## 3. Results

Table 1 provides a descriptive summary of the study population, consisting of 85 PLHIV with diarrhea. Among these patients, 18.8% (16 out of 85) tested positive for either *E. bieneusi* or *E. intestinalis*, while the remaining 81.2% (69 out of 85) tested negative. Specifically, 16.5% (14 out of 85) tested positive for *E. bieneusi*, and 2.4% (2 out of 85) were positive for *E. intestinalis*. The distribution of infections was statistically significant (*p* < 0.001 for both *E. bieneusi* and *E. intestinalis*).

In terms of age categories, 47.1% of participants were between 18 and 39 years, 47.1% were between 40 and 59 years, and 5.9% were 60 years or older. Participants aged 60+ were more likely to be infected (12.5%) compared to younger age groups (37.5% for 18–39 years and 50.0% for 40–59 years), though the differences were not statistically significant (*p* = 0.389). Regarding gender, the majority of participants were male (78.8%), with females comprising 21.2%. However, females showed significantly higher infection rates (43.8% vs. 15.9%, *p* = 0.014).

Marital status showed that 72.9% of participants were never married, while 27.1% had been married at some point. A slightly higher proportion of infections occurred among participants who had never married (81.2% vs. 18.8%), though the differences were not statistically significant (*p* = 0.406). Educational levels indicated that 62.4% of participants had completed primary education or less, while 37.6% had attained secondary education or higher. Interestingly, infections were more common among those with higher education levels (50.0% vs. 34.8% in the primary group), but this difference was not significant (*p* = 0.258).

Adherence to antiretroviral therapy (ART) showed that 54.1% were non-adherent, 21.2% were continuous adherents, and 24.7% had started ART recently. Infection rates were slightly higher among non-adherent participants (62.5% infected vs. 52.2% non-infected), emphasizing the potential importance of adherence, though this finding was not statistically significant (*p* = 0.620).

Viral load categories showed that 55.3% of participants had undetectable/low viral loads (<1000 copies/mL), 3.5% had moderate viral loads (1000–9999 copies/mL), and 41.2% had high/very high viral loads (≥10,000 copies/mL). Participants with moderate viral loads had a higher proportion of infections (6.2%) compared to those with high/very high viral loads (37.5%), though this difference was not statistically significant (*p* = 0.786).

CD4 counts were categorized as severely low (<200 cells/mm^3^) in 69.4% of participants, moderate (200–349 cells/mm^3^) in 14.1%, and normal/high (≥350 cells/mm^3^) in 16.5%. Infections were more common among participants with normal/high CD4 counts (18.8%) compared to those with moderate (12.5%) or severely low CD4 counts (68.8%), though the differences were not statistically significant (*p* = 0.951).

Employment status showed that 70.6% of participants were unemployed, while 29.4% were employed, with slightly higher infection rates among the unemployed (75.0% vs. 25.0%), though not statistically significant (*p* = 0.667). Substance use showed significant differences between groups (*p* = 0.039), with a higher proportion of substance users in the uninfected group (53.6% vs. 25.0%).

The logistic regression analysis demonstrated that certain factors are associated with the likelihood of being positive for *E. bieneusi* or *E. intestinalis* (Table 2). Age was significantly associated with an increased likelihood of infection for participants aged 60+ compared to those aged 18–39 (OR = 19.169, *p* < 0.05), suggesting that older individuals had substantially higher odds of infection. However, the association for the 40–59 age group compared to the 18–39 group was not statistically significant (OR = 1.567, *p* = 0.560).

Gender was significantly associated with infection, with females having higher odds of infection compared to males (OR = 10.491, *p* < 0.05). Marital status showed no strong evidence of association, as participants who had never been married had higher odds of infection compared to those who had been married (OR = 4.764, *p* = 0.113), but the result was not statistically significant.

Education level, comparing those with secondary education or higher to those with primary education or lower, was not statistically significant (OR = 3.480, *p* = 0.137). ART adherence was marginally significant; non-adherent participants had increased odds of infection compared to those who were continuously adherent (OR = 8.466, *p* < 0.10). Participants who had recently started ART showed no significant association with infection (OR = 2.413, *p* = 0.464).

Viral load was marginally associated with infection for participants with high/very high viral loads (≥10,000 copies/mL) compared to those with undetectable/low viral loads (<1000 copies/mL) (OR = 0.457, *p* < 0.10), suggesting a potential protective effect. However, participants with moderate viral loads (1000–9999 copies/mL) did not show a significant association (OR = 0.127, *p* = 0.314).

CD4 count categories showed no significant associations with infection. Participants with moderate CD4 counts (200–349 cells/mm^3^) had lower odds of infection compared to those with severely low CD4 counts (<200 cells/mm^3^) (OR = 0.712, *p* = 0.743), while those with normal/high CD4 counts (≥350 cells/mm^3^) also had reduced odds of infection (OR = 0.064, *p* = 0.130), though these results were not statistically significant.

Employment status was not significantly associated with infection, as participants who were employed had similar odds of infection compared to those who were unemployed (OR = 1.048, *p* = 0.963). Substance use was marginally associated with lower odds of infection (OR = 0.262, *p* < 0.10), indicating a potential protective effect.

## 4. Discussion

This study provides the first systematic analysis of *Enterocytozoon bieneusi* and *Encephalitozoon intestinalis* infections among PLHIV with diarrhea in Ecuador, revealing a prevalence of 18.8%. The logistic regression analysis identified significant predictors such as age, sex, and ART adherence, underscoring critical factors influencing susceptibility to microsporidia infections in this vulnerable population.

The observed prevalence in Ecuador was notably higher than in other regions, such as the Republic of Korea, where *E. intestinalis* and *E. bieneusi* were detected in 1.93% and 0.08% of patients with acute diarrhea, respectively [32]. Similarly, in India, *E. bieneusi* was found in 1.8% of PLHIV with CD4+ cell counts ≤100 cells/µL [33]. These differences highlight the influence of environmental, genetic, or socio-economic factors, suggesting the need for region-specific research and interventions to address these disparities. Future studies using advanced technologies like genomics and proteomics could provide deeper insights into pathogen–host interactions in Ecuador’s unique context.

Participants aged 60+ years had significantly higher odds of infection compared to those aged 18–39 years (OR = 19.169, *p* = 0.036), consistent with research suggesting immune senescence increases vulnerability to opportunistic infections [34]. Targeted interventions, such as routine monitoring and tailored preventive strategies for older PLHIV, could mitigate this risk. However, findings for the 40–59 age group were not significant, emphasizing the need for further exploration of age-related patterns in infection susceptibility.

Sex was significantly associated with infection, with females having 10 times higher odds compared to males (OR = 10.491, *p* = 0.028). This aligns with studies that suggest sex differences in immune response and health-seeking behaviors may influence infection risk. However, the small number of female participants necessitates cautious interpretation and further validation in larger, more balanced cohorts.

ART adherence emerged as a crucial factor, with non-adherent participants showing marginally higher odds of infection (OR = 8.466, *p* = 0.077). This aligns with findings from Colombia, where non-adherence increased the risk of opportunistic infections, including microsporidia [35,36]. Improving ART adherence through targeted support programs addressing socioeconomic and psychological barriers could significantly reduce infection rates [37].

Participants with high/very high viral loads (≥10,000 copies/mL) showed marginally reduced odds of infection compared to those with undetectable/low viral loads (OR = 0.457, *p* = 0.094). Although higher viral loads are typically associated with increased susceptibility to infections, this result could also reflect unmeasured confounders, such as healthcare-seeking behaviors, ART history, or variations in diagnostic testing. Further studies with larger samples are needed to clarify this association.

Substance use showed a marginally significant inverse association with infection (OR = 0.262, *p* = 0.094). While counterintuitive, this result could be influenced by factors such as increased healthcare engagement among substance users or differences in immune response. Previous studies have linked substance use to poorer health outcomes in PLHIV [8], and further research is needed to explore potential confounders and behavioral factors.

Although no significant associations were found for CD4 count categories, descriptive trends indicated lower odds of infection for participants with normal/high CD4 counts (≥350 cells/mm^3^) compared to those with severely low counts (<200 cells/mm^3^). These findings reinforce the importance of maintaining immune function through ART to reduce infection risk.

Despite these findings, the study’s limitations must be acknowledged. The single-center design and small sample size (n = 85), including only 16 positive cases, may reduce statistical power and increase the risk of type II errors [38,39]. These limitations could affect the robustness of some findings and restrict generalizability to broader populations of PLHIV. Future multi-center studies with larger sample sizes are essential to validate these results and identify additional risk factors, such as socioeconomic status, mental health, and detailed substance use patterns.

The zoonotic potential of *E. bieneusi* and *E. intestinalis*, as suggested by studies in children and animals, also warrants further investigation in Ecuador [36]. Understanding these zoonotic links through personalized genomic and proteomic analyses could provide insights into broader public health implications and aid in developing strategies to prevent cross-species transmission. Precision public health efforts integrating such tools could strengthen infection prevention and management strategies.

This study underscores the importance of ART adherence and public health strategies to mitigate microsporidia infection risk among PLHIV. In Ecuador, where prevalence among PLHIV with diarrhea has been reported at 25% [19], targeted interventions that incorporate regular monitoring of viral load, ART optimization, and tailored patient support programs are critical. Integrating cost-effective, precision medicine approaches could further enhance outcomes for this population. Advanced diagnostic techniques, as used in studies from Slovakia [40] and Argentina [41], and omics-based strategies [42], could provide new avenues for understanding and managing microsporidia infections.

## 5. Conclusions

This study provides important insights into the prevalence and associated risk factors for *E. bieneusi* and *E. intestinalis* infections among PLHIV with diarrhea in Ecuador, identifying significant predictors such as age, ART adherence, and HIV viral load. While the findings highlight the potential for targeted public health interventions and personalized approaches to reduce the burden of microsporidia infections, the study’s limitations must be acknowledged. These include the small sample size, single-center design, and exploratory nature of some analyses, which warrant caution in generalizing the results. Despite these limitations, the study offers a valuable foundation for future research and the development of precise diagnostic and treatment strategies tailored to individual patient needs. This work contributes to the broader understanding of microsporidia infections and underscores the importance of integrating personalized medicine into public health efforts.

## Figures and Tables

**Figure 1 jcm-14-00348-f001:**
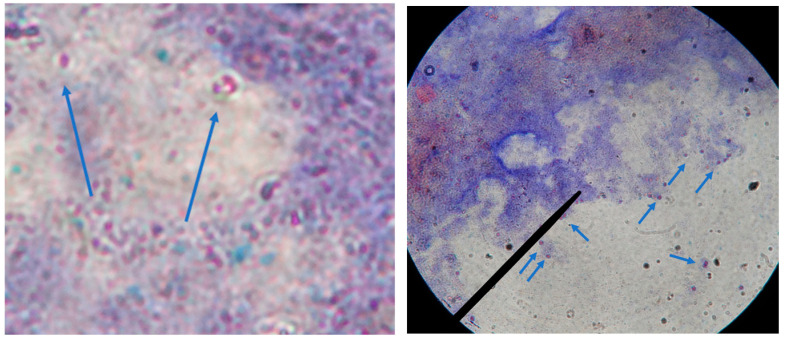
Chromotropic Gram stain of microsporidia observed under light microscopy at 1000× magnification. The blue arrows indicate the spores of microsporidia, characterized by their ovoid shape and distinctive purple staining. These spores are surrounded by a faintly stained background, aiding in their identification.

**Figure 2 jcm-14-00348-f002:**
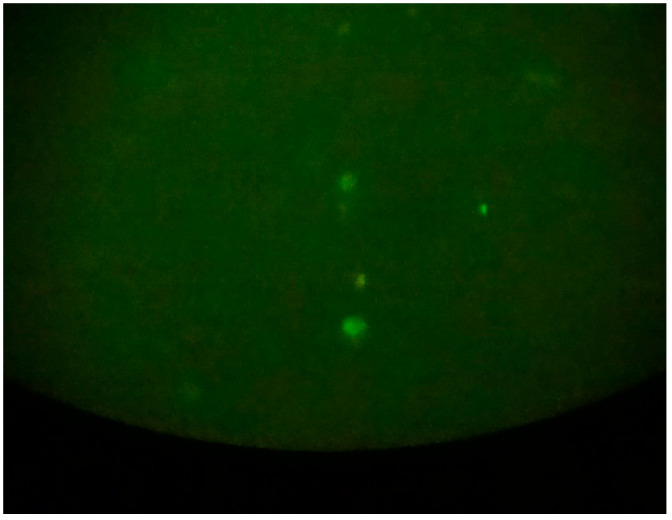
*E. bieneusi* observed using immunofluorescence technique with monoclonal antibodies at 1000× magnification.

**Figure 3 jcm-14-00348-f003:**
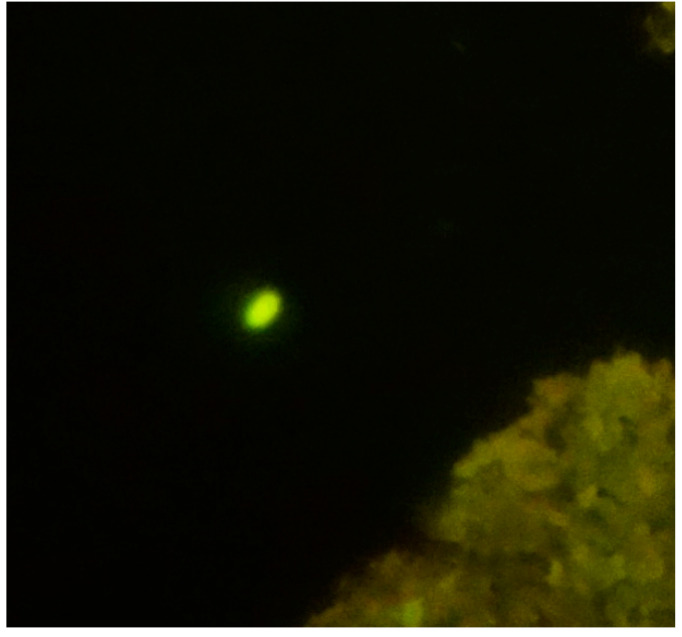
*E. intestinalis* observed using immunofluorescence technique with monoclonal antibodies at 1000× magnification.

**Figure 4 jcm-14-00348-f004:**
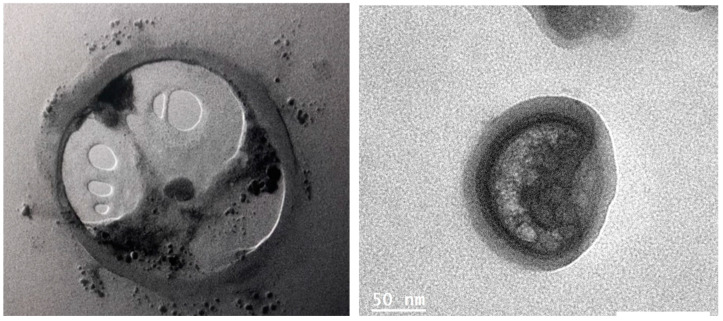
*E. bieneusi* observed under transmission electron microscopy (TEM) at 60,000× magnification, 60 kV.

**Figure 5 jcm-14-00348-f005:**
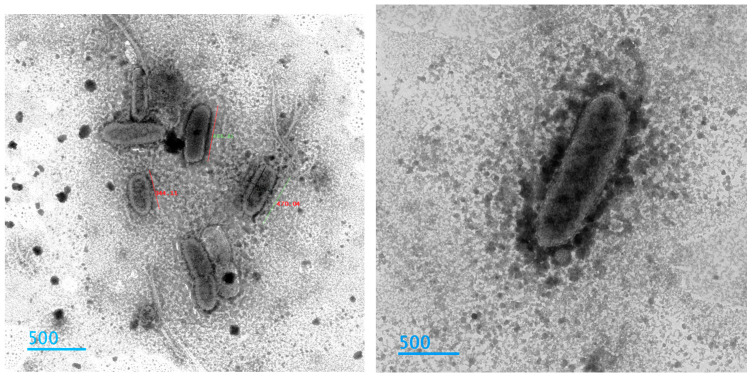
*E. intestinalis* observed under transmission electron microscopy (TEM) at 60,000× magnification, 60 kV.

**Table 1 jcm-14-00348-t001:** Descriptive statistics by microsporidia infection status.

Variable	Category	Negative (n = 69)	Positive (n = 16)	Total (n = 85)	*p*-Value
Microsporidia infection status (*E. bieneusi* or *E. intestinalis*)	Negative	69 (81.2%)	0 (0.0%)	69 (81.2%)	<0.001
	Positive	0 (0.0%)	16 (18.8%)	16 (18.8%)	
*E. bieneusi*	Negative	69 (100.0%)	2 (12.5%)	71 (83.5%)	<0.001
	Positive	0 (0.0%)	14 (87.5%)	14 (16.5%)	
*E. intestinalis*	Negative	69 (100.0%)	14 (87.5%)	83 (97.6%)	0.003
	Positive	0 (0.0%)	2 (12.5%)	2 (2.4%)	
Age Category	18–39	34 (49.3%)	6 (37.5%)	40 (47.1%)	0.389
	40–59	32 (46.4%)	8 (50.0%)	40 (47.1%)	
	60+	3 (4.3%)	2 (12.5%)	5 (5.9%)	
Sex	Male	58 (84.1%)	9 (56.2%)	67 (78.8%)	0.014
	Female	11 (15.9%)	7 (43.8%)	18 (21.2%)	
Marital Status	Never Married	49 (71.0%)	13 (81.2%)	62 (72.9%)	0.406
	Married	20 (29.0%)	3 (18.8%)	23 (27.1%)	
Education Level	Primary	45 (65.2%)	8 (50.0%)	53 (62.4%)	0.258
	Secondary+	24 (34.8%)	8 (50.0%)	32 (37.6%)	
ART Adherence	Non-Adherent	36 (52.2%)	10 (62.5%)	46 (54.1%)	0.62
	Continues	16 (23.2%)	2 (12.5%)	18 (21.2%)	
	Started Recently	17 (24.6%)	4 (25.0%)	21 (24.7%)	
Viral Load	Undetectable/Low (<1000 copies)	38 (55.1%)	9 (56.2%)	47 (55.3%)	0.786
	Moderate (1000–9999 copies)	2 (2.9%)	1 (6.2%)	3 (3.5%)	
	High/Very High (≥10,000 copies)	29 (42.0%)	6 (37.5%)	35 (41.2%)	
CD4 Count	Severely Low (<200)	48 (69.6%)	11 (68.8%)	59 (69.4%)	0.951
	Moderate (200–349)	10 (14.5%)	2 (12.5%)	12 (14.1%)	
	Normal/High (≥350)	11 (15.9%)	3 (18.8%)	14 (16.5%)	
Employment	No	48 (69.6%)	12 (75.0%)	60 (70.6%)	0.667
	Yes	21 (30.4%)	4 (25.0%)	25 (29.4%)	
Substance Use	No	32 (46.4%)	12 (75.0%)	44 (51.8%)	0.039
	Yes	37 (53.6%)	4 (25.0%)	41 (48.2%)	

Note: All variables in the table are categorical, and the *p*-values correspond to Chi-square tests for associations between each variable and microsporidia infection status.

**Table 2 jcm-14-00348-t002:** Analysis of factors associated with microsporidia infection (*E. bieneusi* or *E. intestinalis*) status in PLHIV (n = 85) through logistic regression.

Variable	Category	Odds Ratio (OR)	95% CI	*p*-Value
Age (Ref = 18–39)	40–59	1.567	0.346–7.110	0.56
	60 ^+^	19.169 *	1.221–300.886	0.036
Sex (Ref = Male)	Female	10.491 *	1.286–85.553	0.028
Marital Status (Ref = Married)	Never Married	4.764	0.690–32.901	0.113
Education Level (Ref = Primary or Lower)	Secondary ^+^	3.48	0.673–17.998	0.137
ART Adherence (Ref = Continues)	Non-Adherent	8.466 ^+^	0.792–90.463	0.077
	Started Recently	2.413	0.228–25.506	0.464
Viral Load (Ref = Undetectable/Low)	Moderate (1000–9999 copies/mL)	0.127	0.002–7.065	0.314
	High/Very High (≥10,000 copies/mL)	0.457 ^+^	0.066–3.175	0.094
CD4 Count (Ref ≤ 200)	Moderate (200–349)	0.712	0.093–5.432	0.743
	Normal/High (≥350)	0.064	0.002–2.252	0.13
Employment (Ref = No)	Yes	1.048	0.144–7.646	0.963
Substance Use (Ref = No)	Yes	0.262 ^+^	0.055–1.256	0.094
Intercept		0.012 *	0.0004–0.299	0.007

*p* values: * *p* < 0.05, ^+^ *p* < 0.10.

## Data Availability

The data presented in this study are available on request from the corresponding author due to the vulnerability of the PLHIV involved.

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
