# Peer review of "Prevalence and Individualized Risk Factors of E. bieneusi and E. intestinalis Infections Among People Living with HIV (PLHIV) with Diarrhea in Ecuador: Insights from a Single-Center Cross-Sectional Study"

_jcm, 2025, doi:10.3390/jcm14020348_

Round 1
Reviewer 1 Report
Comments and Suggestions for Authors
The authors have treated a topic that is not common in the general HIV-1 literature. Among publications about opportunistic infections, this is the first I have seen on microsporidia. The paper is generally well researched, the study design makes good sense, the methods appear to be carried out well and the results are generally interesting.
I do have a number of questions, mostly about methods and the reporting of results. Let me take them in order.
In the abstract (line 26), you initiate use of the term "substance use" without ever specifying what substances you mean. I think we all can infer that you are discussing recreational drugs, especially intravenous drugs, and their abuse rather than use. But, you should use a different term and completely define it somewhere in the Introduction.
In lines 65 - 72, you establish the prevalence rates for microsporidia in other countries in the range of 12% to 50%. However, in lines 205 - 207, you state that your sample's prevalence of 18.8% is "significant". How did you determine that a value right in median area (not exact) is significant, which implies that it is out of the ordinary? Needs an explanation.
On line 91, you offer two references to support the idea that meeting the unique healthcare needs of these patients are critical ethical research practices. Reference 24 is very specific to India and could be replaced with another more generally applicable reference. I don't see the relevance of reference 25 to the paper in this context.
On line 113 you mention the "inclusion criteria", but you don't make them specific. You need to do that either in the text or in a supplemental exhibit.
On line 195, you state that the presence of either species of microsporidia would constitute infection. Did you do any analysis in which the two were treated separately? At this point in the paper, we the readers have no idea if these species are alike in effect (you indicate differences in structure at a superficial level). Some differentiation in how they associate with the risk factors would be very interesting and I suspect you have that data as stated on line 160.
The figures were very helpful to understand the structure of the microsporidia.
lines 169 - 173: I found the variables Sex and Marital Status not entirely useful to understanding risks to people living with HIV-1 (PLWH). Gender has become a fluid concept in recent years and of greater interest to HIV research than biological gender is the patient's sexual behavior, which defines in large part the risk of infection and the risk of contracting an opportunistic infection that is associated with the gut (as evidenced by your using stool samples as the research material).
line 177: HIV Viral Load. You have used raw copies per mL. It is more common today to use the log base 10 transformation of the raw copies as it gives an easier to comprehend number. I recommend you change this Table 1.
Table 1: This needs some restructuring. It's not really clear. I understand your characterization of categorical variables with a number and a percentage of total. However, what are the numbers for continuous variables. I assume they are mean and standard deviation. But you need to make that clear.
I would also back Table 1 up with some graphs showing visually how age or viral load compare to sex or adherence. Even better if STATA lets you show whether the patient was positive or negative for the microsporidia.
In populations we study of general HIV-1 patients, we generally find average viral load is low and CD4 is higher than you found. In this group, the average CD4 level is below 200, indicating that the average of your group is already at risk for conversion to AIDS. Because of the high standard deviation (if that's what it is), this is a sample with an extremely high coefficient of variation. This in itself should be spoken of, why it may have come about and show a graph comparing these levels to adherence. It would be worthwhile as well to invest some time in looking for potential confounders that are creating these really bad values. The table also needs a caption.
Table 2. Again, this needs some work. First, the stars in your variables are not consistent with the legend at the bottom. You refer to a variable as the CONSTANT. I think you are referring to the Intercept of the regression, which is the OR if all other OR's are 0. But, check with your STATA manual as to what it thinks the value means and make this clear. The OR of HIV Viral Load is 1.0, but has a statistically significant p-value. You should give that an interpretation. I think it is that HIV-1 levels are not related to the presence or absence of microporidia.
Line 288: why is substance (ab)use not related to microsporidia as you say. I would think there are a number of behavioral variables that interfere in this association and mask what is going on if you only refer to HIV-1 as variable.
Discussion: In general, this section was quite good and you focused on what the paper has done and not been able to do. I would only raise 2 points.
First, you refer to personalized medicine a lot in the Discussion. We are working on two projects now that make use of personalized medicine in another Latin country with limited public resources. I would be very specific about what kinds of tests and treatments you would suggest be personally designed for a given patient. Doing the necessary screening of HLA profiles, subtyping of the HIV virus and what you need to do to capture the information necessary about the microsporidia species the patient carries could be very expensive in time and money.
On line 333, you refer to your "small" sample and its power. You need to tell us what the power of your study was. Also, 85 is not a small study in a resource limited setting. I'm not sure you need this as a reason for any problems with the study.
In fact, the study, with the caveats I've listed here is solid. My suggestions go more to the format of the presentation.
Comments on the Quality of English LanguageGenerally fine.
Reviewer 2 Report
Comments and Suggestions for Authors
Thanks for allowing me to review this important work. I have a few comments that would strengthen the manuscript. 1. Please change HIV patients to persons living with HIV. It is important not to describe a person as a disease. 2. Please order the keywords alphabetically. 3. The manuscript was written by two people. One wrote in past tense, the other in present tense. Beginning with the results, the English needs editing that I did not see in the front part of the manuscript. Please go through the entire manuscript and change it to past tense. 4. It is unclear how many participants were positive for E bieneusi only or E. Intestinalis only, or were they always positive for both? 5. None of the Figures tell us what the magnification was. 6. The fecal sample collection and storage process is not described. 7. The staining procedures are not described. 8. The discussion does not go into detail about the alarming low adherence rates and whether optimizing treatment and improving viral suppression would lower the rates of both E. bieneusi and E. intestinalis infections significantly.

The results, discussion and conclusion need to be converted to past tense.
